# Percutaneous transhepatic or endoscopic ultrasound-guided biliary drainage in malignant distal bile duct obstruction using a self-expanding metal stent: Study protocol for a prospective European multicenter trial (PUMa trial)

Daniel Schmitz[1,2]*, Carlos T. Valiente[1], Markus Dollhopf[3], Manuel Perez-Miranda[4], Armin Küllmer[5], Joan Gornals[6], Juan Vila[7], Jochen Weigt[8], Torsten Voigtländer[9], Eduardo Redondo-Cerezo[10], Thomas von Hahn[11], Jörg Albert[12], Stephan vom Dahl[13], Torsten Beyna[14], Dirk Hartmann[15], Franziska Franck[3], Francisco Javier García-Alonso[4], Arthur Schmidt[5], Albert Garcia-Sumalla[6], Amaia Arrubla[7], Markus Joerdens[13], Tobias Kleemann[16], José Ramón Aparicio Tomo[17], Felix Grassmann[18], Jochen Rudi[1]

1 Department of Gastroenterology, Oncology and Diabetology, Theresienkrankenhaus und St. Hedwig-Klinik, Lehrkrankenhaus der Universität Heidelberg, Mannheim, Germany, 2 Department of Gastroenterology and Infectiology, Helios Kliniken Schwerin, Schwerin, Germany, 3 Department of Gastroenterology and Hepatology, München Klinik Neuperlach, München, Germany, 4 Department of Gastrointestinal Endoscopy, Hospital Universitario Río Hortega, Valladolid, Spain, 5 Department of Medicine II: Gastroenterology, Hepatology, Endocrinology, and Infectious Diseases, Universitätsklinikum Freiburg, Freiburg, Germany, 6 Department of Digestive Diseases, Hospital Universitari de Bellvitge, Barcelona, Spain, 7 Endoscopy Unit, Complejo Hospitalario de Navarra, Pamplona, Spain, 8 Department of Gastroenterology, Hepatology and Infectiology, Universitätsklinikum Magdeburg, Magdeburg, Germany, 9 Department of Gastroenterology, Hepatology and Endocrinology, Medizinische Hochschule Hannover, Hannover, Germany, 10 Department of Gastrointestinal Endoscopy, Hospital universitario Virgen de las Nieves, Granada, Spain, 11 Department of Gastroenterology, Hepatology, and Interventional Endoscopy, Asklepios Klinik Barmbek, Hamburg, Germany, 12 Department of Gastroenterology, Hepatology and Endocrinology, Robert-Bosch-Krankenhaus, Stuttgart, Germany, 13 Department of Gastroenterology, Hepatology and Infectiology, Universitätsklinikum Düsseldorf, Düsseldorf, Germany, 14 Department of Internal Medicine, Gastroenterology and Interventional Endoscopy Evangelisches Krankenhaus Düsseldorf, Düsseldorf, Germany, 15 Department of General Internal Medicine, Diabetology, Gastroenterology and Oncology, Katholisches Klinikum Mainz, Mainz, Germany, 16 Department of Gastroenterology and Rheumatology, Carl-Thiem-Klinikum Cottbus, Cottbus, Germany, 17 Department of Digestive Diseases, Hospital General Universitario de Alicante, Alicante, Spain, 18 Institute for Medical Statistics and Epidemiology, Medical School Hamburg, Hamburg, Germany

* Daniel.Schmitz@helios-gesundheit.de

**Data Availability Statement:** The data of this study NCT03546049 will be publicly available on the

## Abstract

### Background

Endoscopic ultrasound-guided biliary drainage (EUS-BD) was associated with better clinical success and a lower rate of adverse events (AEs) than fluoroscopy-guided percutaneous transhepatic biliary drainage (PTBD) in recent single center studies with mainly retrospective design and small case numbers (< 50). The aim of this prospective European multicenter study is to compare both drainage procedures using ultrasound-guidance and primary metal stent implantation in patients with malignant distal bile duct obstruction (PUMa Trial).

public repository FAIRsharing ID 4350; https://
fairsharing.org.

**Funding:** The author(s) received no specific
funding for this work

**Competing interests:** The authors have declared
that no competing interests exist.

## Methods

The study is designed as a non-randomized, controlled, parallel group, non-inferiority trial. Each of the 16 study centers performs the procedure with the best local expertise (PTBD or EUS-BD). In PTBD, bile duct access is performed by ultrasound guidance. EUS-BD is performed as an endoscopic ultrasound (EUS)-guided hepaticogastrostomy (EUS-HGS), EUS-guided choledochoduodenostomy (EUS-CDS) or EUS-guided antegrade stenting (EUS-AGS). Insertion of a metal stent is intended in both procedures in the first session. Primary end point is technical success. Secondary end points are clinical success, duration pf procedure, AEs graded by severity, length of hospital stay, re-intervention rate and survival within 6 months. The target case number is 212 patients (12 calculated dropouts included).

## Discussion

This study might help to clarify whether PTBD is non-inferior to EUS-BD concerning technical success, and whether one of both interventions is superior in terms of efficacy and safety in one or more secondary endpoints. Randomization is not provided as both procedures are rarely used after failed endoscopic biliary drainage and study centers usually prefer one of both procedures that they can perform best.

## Trial registration

ClinicalTrials.gov ID: NCT03546049 (22.05.2018).

## Introduction

In malignant distal bile duct obstruction, endoscopic retrograde cholangiopancreatography (ERCP) is the method of first choice for biliary drainage [1]. Very rarely, ERCP is not successful or not possible due to anatomical changes [2]. In these cases, bile duct must be approached percutaneously or transluminal by endoscopic ultrasound-guidance. A recently published meta-analysis on PTBD versus EUS-BD recommended EUS-BD as the preferred method when ERCP fails as EUS-BD showed a higher clinical success rate, less AEs, and a lower rate of re-interventions [3]. However, the included six studies (Table 1) were mostly single center studies [4, 6–8], mainly retrospective [4, 6, 8, 9], and had predominantly small numbers of cases (n = 25–73). Furthermore, the used PTBD technique in the included studies often was not equivalent to the used EUS-BD technique. For example, plastic stents were used in PTBD although only metal stents were inserted in EUS-BD in two studies [4, 5], fluoroscopic guidance was only used for bile duct access in several studies [4, 5], benign and malignant diseases were mixed [6] and PTBD had many scheduled re-interventions [4, 5, 7–9]. On the other side, only one [4] of six studies compared all three relevant EUS-BD methods that are used in real world-setting with EUS-BD. These are EUS-guided choledochoduodenostomy (EUS-CDS), EUS-guided hepaticogastrostomy (EUS-HGS) and EUS-guided antegrade stenting (EUS-AGS). Both the historically "older" technique of PTBD [10] as well as the historically "younger" technique of EUS-BD [11] might be associated with different AEs of different severity grades.

In PTBD, recent studies had shown that the AE rate can be reduced by ultrasound-guided bile duct access instead of pure fluoroscopy guidance [12], left-sided transhepatic bile duct

**Table 1. Commented overview of the reported PTBD procedures in the published comparative studies on PTBD versus EUS-BD.**

| Author + year | Study type | PTBDs (n) | Adverse events (n) | Re-intervent-ions/patient | Guidance of bile duct access | Further comments | Compared EUS-BD method |
|---|---|---|---|---|---|---|---|
| Artifon 2012 [7] | Prosp. Single center | 12 | 3/12 (25%) | Not reported | Fluoroscopy Ultrasound | 4 external drainages before metal stent insertion | EUS-CDS |
| Bapaye 2013 [4] | Retro. Single Center | 26 | 12/26 (46%) | Not reported | Fluoroscopy | Only 12/26 (46%) metal stents and 14/26 (54%) external drainages | EUS-CDS |
| | | | | | | | EUS-HGS |
| | | | | | | | EUS-AGS |
| Kashab 2015 [8] | Retro. Single Center | 51 | 20/51 (39%) | 0.80 | Not reported | Not reported whether metal stents were used or not, many scheduled re- interventions, many bile leaks (n = 17) | EUS-CDS |
| | | | | | | | EUS-RV |
| Sharaiha 2016 [6] | Retro. Single Center | 13 | 7/13 (54%) | 1.70 | Not reported | Benign and malignant bile duct obstruction were mixed, number of metal stents remains unclear | EUS-HGS |
| | | | | | | | EUS-CDS |
| Lee 2016 [5] | Prosp. 4 Centers | 32 | 10/32 (31%) | 0.93 | Fluoroscopy | 2 step-intervention: external drainage before metal stent insertion, only 15/32 (48%) metal stents inserted | EUS-CDS |
| | | | | | | | EUS-HGS |
| Sportes 2017 [9] | Retro. 2 Centers | 20 | 2/20 (10%) | 1.05 | Ultrasound | External drain was left after metal stent implantation and removed some days later when stent implantation was clinically successful, scheduled re-interventions were mixed with unscheduled re-interventions | EUS-HGS |

access distant to the pleural space [13], primary metal stent implantation without remaining external drainage catheter [14] or portal vein-oriented access to the peripheral intrahepatic bile duct if the bile duct is hardly dilated [15]. In EUS-BD, AEs such as biliary leak or stent migration might be avoided by a newer generation of inserted metal stents such as partially covered metal stents in EUS-HGS [16] and lumen apposing metal stents (LAMS) in EUS-CDS [17].

In conclusion, there was an unmet need to prospectively compare PTBD with EUS-BD on an updated level of standard care considering optimal bile duct access, appropriate metal stents, timing of metal stent implantation, EUS-BD techniques, and documentation of AEs according to severity grades in a well-defined disease entity. Therefore, this prospective European multicenter study (PUMa) with 16 study centers and a target case number of n = 212 was initiated to compare PTBD with EUS-BD (EUS-CDS, EUS-HGS and EUS-AGS) in patients with malignant, unresectable, distal bile duct obstruction when ERCP has failed (Clinical-Trials.gov ID: NCT03546049).

## Materials and methods

This is a prospective, multicenter, non-randomized, open, confirmatory, 2-arm non-inferiority study with a parallel group design. It will include at least 106 patients per study arm with a total number of 212 patients and with a calculated number of dropouts of 12 patients. Patients will be recruited from 16 community and academic hospitals (EUS-BD: n = 8, PTBD: n = 8) in Spain and Germany (S1 File). The first patient was enrolled on 28th December 2018. The course of the study from screening until the end of follow-up after 6 months as well as the corresponding control visits are shown in a SPIRIT schedule (Fig 1).

### Primary endpoint

Technical success, defined as the successful implantation of a metal stent into the bile duct system confirmed by injection of contrast medium.

| TIMEPOINT* | Enrolment | Allocation | Post-allocation | | | |
|---|---|---|---|---|---|---|
| | -t1 | 0 | t1 | t2 | t3 | t4 |
| **ENROLMENT:** | | | | | | |
| *Eligibility screen* | X | | | | | |
| *Informed consent* | X | | | | | |
| *Allocation* | | X | | | | |
| **INTERVENTIONS:** | | | | | | |
| *EUS-BD* | | | X | | | |
| *PTBD* | | | X | | | |
| **ASSESSMENTS:** | | | | | | |
| *Bilirubin value mg/dl* | X | | | | X | |
| *Pain sensation (VAS)* | X | | | X | X | |
| *Technical success of intervention* | | | X | | | |
| *Duration of intervention* | | | X | | | |
| *Reintervention* | | | | X | X | X |
| *Length of hospital stay* | | | | | X | |
| *Adverse events* | | | | X | X | X |
| *Disease specific survival* | | | | | | X |
| *Overall survival* | | | | | | X |

**Fig 1. SPIRIT schedule.** * List specific timepoints in this row: -t$_1$ = one day before intervention, t$_1$ = day after intervention, t$_2$ = one day after intervention, t$_3$ = 7 days after intervention, t$_4$ = 6 months after intervention.

## Secondary endpoints

- Clinical success, defined as the reduction of the serum bilirubin value of $\geq$ 50% in 7 days

- Occurrence of AEs (including death) within 30 days after the intervention

- Duration of the biliary drainage procedure in minutes

- Patient reported pain on day 1 and day 7 after the intervention, documented by Visual Analogue Scale (VAS)

- Duration of hospital stay in days

- Number of re-interventions from the day of the first intervention to the end of the follow-up

- Overall survival and disease-specific survival in days within the follow-up of six months

**Inclusion criteria.**

- Patients ≥ 18 years of age with inoperable malignant distal bile duct obstruction (including tumors at the biliodigestive anastomosis) in whom ERCP was not successful or was not possible due to surgical anatomical changes

- At least a twofold increase in the serum bilirubin value = bilirubin value ≥ 2. 0 mg/dl

- The underlying malignant disease must be verified histologically

- Abdominal sonography and a further cross-sectional imaging (CT and/or MRI) must be performed to rule out extended malignant disease with an estimated survival time of less than 3 months (e.g., due to extensive liver metastases)

- Written informed consent by the patient

**Exclusion criteria.**

- Limited blood clotting: Quick < 50%, PTT > 50 sec., platelet count < 50/nl

- Extrahepatic cholangiocarcinoma of the Klatskin type in stage Bismuth 2–4 or intrahepatic cholangiocarcinoma

- Proximal malignant extrahepatic bile duct obstruction defined as obstruction at the level of the bifurcation of the extrahepatic bile duct and proximal of it

- Operable carcinoma or disease that can be cured by drug-based oncological therapy (e.g., aggressive non-Hodgkin's lymphoma)

- Pregnancy or breastfeeding women

- Patients participating in another study on PTBD or EUS-BD

As both PTBD and EUS-BD are complex and less frequent "second-line" interventions in distal malignant bile duct obstruction even in high volume centers, centers usually prefer one of both interventions to deliver the highest quality of treatment for their patients. Therefore, it was consciously decided not to randomize the patients. Due to the nature of the biliary tract interventions, the patient and the treating physician cannot be blinded.

## Interventions

Both PTBD as well as EUS-BD are performed by interventional gastroenterologists with experience of > 8 years in performing the intervention.

**PTBD.** The technique of the PTBD has already been described in detail [14]. Ultrasound guidance should be used when the bile duct will be accessed percutaneously (Fig 2). Fluoroscopy is usually necessary to show the bile duct system by injecting a radiopaque contrast medium. If possible, a left-sided bile duct access should be preferred. Metal stent implantation

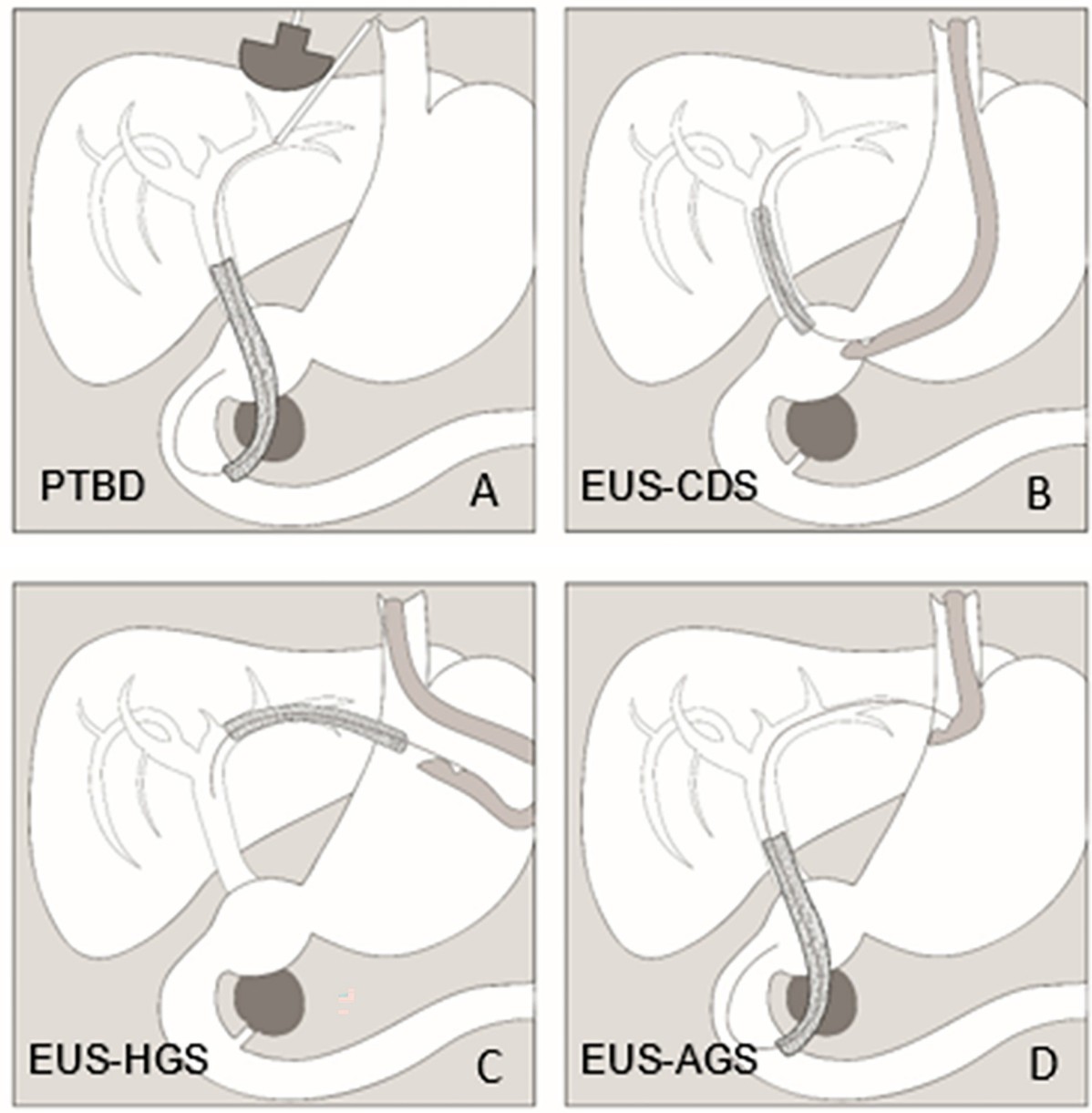

**Fig 2. Illustration of the four biliary drainage procedures in the PUMa trial.** 1A: PTBD with ultrasound-guided bile duct access, 1B: EUS-guided choledochoduodenostomy (EUS-CDS), 1C: EUS-guided hepaticogastrostomy (EUS-HGS), 1D: EUS-guided antegrade stenting (EUS-AGS).

should be performed in the first session without a remaining external catheter but is not mandatory. The choice of the metal stent is up to the preference of the investigator and depends on the patient condition (normally, a diameter of 8–10 mm x length of 60–100 mm is used). The optimal release of the metal stent can be endoscopically controlled under visual inspection of a second investigator if possible.

**EUS-BD.** The techniques of the EUS-BD have already been described in detail [18]. In this study, the EUS-BD procedures comprise EUS-CDS, EUS-HGS and EUS-AGS (Fig 2).

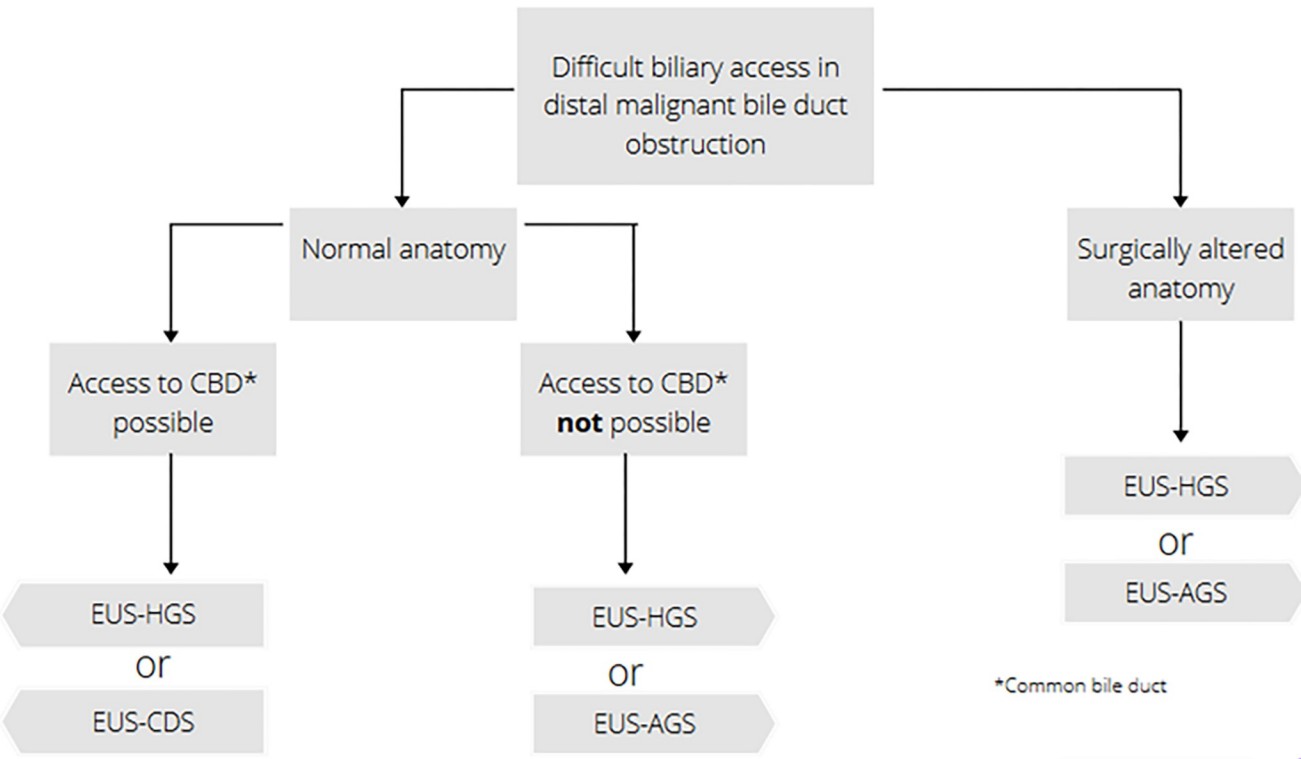

**Fig 3. Algorithm for the selected EUS-BD procedure in distal, malignant bile duct obstruction.**

EUS-BD will be performed according to the algorithm presented in Fig 3. Usually, the metal stent is inserted in the same session. The choice of the metal stent is up to the preference of the investigator and depends on the patient condition. Normally, a diameter of 8–10 mm x length of 60–100 mm is used in EUS-AGS, a LAMS-diameter of 6–8 mm is used in EUS-CDS, and a diameter of 8–10 mm x length of 80–120 mm is used in EUS-HGS.

**Co-interventions.** After clinically successful biliary intervention (normalized serum bilirubin value), a drug-based oncological therapy might be applied depending on the patient's general condition and specific malignant disease.

**Early and late re-interventions.** All interventions that are necessary to achieve biliary drainage or to restore biliary drainage after failed primary successful biliary drainage are defined as re-interventions. According to this definition a renewed or repeated PTBD or EUS-BD, an exchange of a percutaneous external biliary drainage to transpapillary metal stent in PTBD, a rescue cross over biliary drainage procedure after failed EUS-BD or failed PTBD, and a surgical application of a biliodigestive anastomosis are re-interventions. Early re-interventions are defined as re-interventions from the beginning up to 30 days in the follow up. Late re-interventions are defined as re-interventions from 31 days to 6 months in the follow up. Scheduled re-interventions should be strictly avoided in both compared procedures and usually are not necessary after metal stent implantation.

## Follow-up

The study and the corresponding control visits will end for each patient after a follow-up of 6 months after the intervention. Each study center is obliged to document study patients who have met the inclusion criteria but were not included in the study. Excluded patients and

**Table 2. Classification of AEs by severity in line with the recommendations of the ASGE workgroup 2010.**

| Consequence of adverse event | Severity Grade | | | |
|---|---|---|---|---|
| | mild | moderate | severe | fatal |
| Procedure aborted (or not started) because of an adverse event | x | | | |
| Unplanned anesthesia/ventilation support, i.e., endotr. intubation during conscious sedation* | | x | | |
| Unplanned admission or prolongation of hospital stay for up to 3 nights | x | | | |
| Prolongation of hospital stay for 4–10 nights | | x | | |
| Prolongation of hospital stay for > 10 nights | | | x | |
| Intensive care unit admission for 1 night | | x | | |
| Intensive care unit admission for > 1 night | | | x | |
| Blood transfusion | | x | | |
| Further endoscopic or percutaneous transhepatic intervention necessary (e.g., for hemostasis) | | x | | |
| Radiological intervention necessary (e.g., for coiling) | | x | | |
| Surgical intervention necessary | | | x | |
| Permanent damage/permanent disability remains | | | x | |
| Death | | | | x |

*Temporary ventilation support by bagging or nasal airway during conscious sedation, and endotracheal intubation during a modified anesthesia care procedure are not adverse events.

reasons for exclusion will be presented in an adapted CONSORT flow diagram [19]. All adverse events must be systematically recorded and classified according to an adapted AE severity grading system (Table 2) of the American Society of Gastrointestinal Endoscopy (ASGE) [20].

## Data management

All data collected in the study will be documented on standardized case report forms (CRFs). The CRFs must ensure the complete documentation of all patient data to be collected according to the study protocol. Database development, data entry by means of double entry, data management and data validation are the responsibility of the study center of the Principal Clinical Investigator (PCI). All data management processes are carried out in accordance with standard operation procedures (SOPs). The data validation includes checks for completeness, consistency and plausibility of the data documented in the CRF. For this purpose, a query system is established between data management and the investigator. In the so-called query process, queries are sent from data management to the trial center as quickly as possible to clarify incomplete, implausible, and/or inconsistent data. These queries are answered by the investigator, or a person authorized by the investigator and then sent to the data management system for entry into the database. Once all queries for all included patients have been resolved, the database will be closed at the end of the study and handed over to the biometrician for evaluation. After completion of all evaluations and preparation of the final report, all original CRFs will be transferred to the PCI for archiving. At the end of the study, the data will be transformed into various data formats (e.g., CSV files) to enable further use. All primary data are planned to be made publicly available for re- and meta-analysis after the end of the study (FAIRsharing ID 4350; https://fairsharing.org). In accordance with the Medical Professional Code, all important study documents (e.g., CRFs) will be archived for at least 10 years after the end of the examination. The PCI or his or her study assistant/study nurse will be responsible for archiving. The documents should contain the protocol, institutional review board application and approval; patient information and informed consent forms; and CRFs and other

reporting forms as well as the final report. Any change in the ownership of the data will be documented. All data, including patient identification, will be made available to regulatory authorities upon request.

## Safety

For the assessment of patient safety, all relevant AEs within 30 days after intervention will be documented in a separate form (Reporting Form AEs) designed for this purpose. To facilitate documentation, most of the expected intervention-specific AEs are listed in the documentation form, such as bleeding/hematoma, haemobilia, biliary leakage, pneumothorax, (biliary) pleural effusion, pneumoperitoneum, bowl perforation, biliary sepsis, new-onset cholangitis, abscess, (purulent) peritonitis, cholecystitis, and metal stent dysfunction/migration. Reporting of unlisted AEs as well as multiple entries is possible. AEs will be documented by date and grade of severity, which is in line with the classification of the ASGE (Table 2) [20].

The following events do not need to be documented as intervention-related AEs: metal stent occlusion caused by tumor ingrowth or bile duct stones, AEs associated with anesthesia and/or sedation, and AEs associated with chemotherapy. Cholangitis requires special consideration. Only new-onset cholangitis after intervention and not pre-existing cholangitis will be considered an adverse event. Since most patients are treated peri-procedurally by prophylactic antibiotics, cholangitis will only be considered an adverse event if there is a change in antibiotic therapy and if new-onset cholangitis can be safely attributed to the intervention. Post-interventional pain will be documented separately in the standard CRF. Serious undesirable events (SUEs) must be documented and reported separately (Reporting Form SUE). An SUE might occur within the complete course of the study and has one of the following consequences for patients regardless of a secure connection to the intervention carried out: death, unplanned surgical intervention necessary, permanent damage or permanent disability remains, unplanned hospitalization or extended hospital stay > 10 days, or an unplanned stay in the intensive care unit > 1 night. The Reporting Form SUE contains the following information: Identification of the study patient, supervising investigator, description of the SUE: event, start and duration, outcome, causal relationship with the intervention, performed treatment, date and signature of the supervising investigator. Association between intervention and SUE: O proved O probable O possible O unlikely O no O not assessable. Outcome of the SUE: O improvement O not yet restored O permanent damage O fatal, cause of death: O unknown. In the case of an accumulation of SUEs in a study center or trial arm, the Steering Committee will be involved, which can lead to the closure of a study center or the termination of the trial.

## Statistical considerations

**Sample size calculation.** The calculation of the sample size is based on the primary endpoint. According to the current literature, PTBD as well as EUS-BD has reached a technical success rate of up to 95% [5]. A difference in the technical success rates of both interventions of 10% or more is assumed to be clinically relevant. Therefore, the non-inferiority margin is defined as $\delta = 0.1$. With a one-sided significance level of 2.5%, 200 patients (100 per group) are required to perform a Farrington/Manning test with a power of 80% (calculations with PASS 14.0.8). It is assumed that for very few patients the primary endpoint will not be observable, for example, if a severe complication (unrelated to the investigated procedures) requires an early termination of unfinished biliary drainage. To account for this loss in information, at least 106 patients need to be recruited per group to compensate for a dropout rate of 5%.

## Statistical analysis

The primary endpoint is the successful implantation of the self-expanding metal stent in malignant distal bile duct obstruction. The non-inferiority of PTBD to EUS-BD is to be demonstrated. The corresponding test hypothesis is: $H0$: $pEUS\text{-}BD - pPTBD \geq \delta$ vs. $H1$: $pEUS\text{-}BD - pPTBD < \delta$ where pEUS-BD is the technical success rate in the EUS-BD group, pPTBD is the technical success rate in the PTBD group, and $\delta = 0.1$ is the non-inferiority margin. The non-inferiority of PTBD to EUS-BD will be determined with the Farrington/ Manning test with a one-sided significance level of 2.5%. Since this is a non-inferiority study, the primary evaluation is based on both the Intention-To-Treat (ITT) set and the Per-Protocol (PP) set. In the ITT set, all patients in whom the intervention is carried out will be considered, even if the intervention was not technically successful or if participation in the study was terminated prematurely (e.g., due to organ perforation), then this patient would be included in the primary evaluation. For these patients, the intervention will be assessed as "not successfully completed". In the PP set, only patients in whom the intervention was performed per protocol and the primary endpoint could be observed, will be included. For missing values, best-case and worst-case scenarios will be calculated as sensitivity analyses. Additionally, regression models are planned to determine the influence of confounders for the endpoints (e.g., bilirubin value). If non-inferiority is shown in terms of technical success, then the occurrence of at least one adverse event will be compared hierarchically (expected adverse event rate for both interventions: 10–20%) and tested for differences between the two groups (based on the ITT set). Due to the hierarchical approach, the full alpha level will be used: two-sided 5%. Further secondary endpoints will be evaluated in the ITT set. It is assumed that there will be fewer adverse events with PTBD than with EUS-BD. Missing values will not be imputed in the analysis of secondary endpoints and the analysis is purely descriptive. Two group t-tests might be used for variables that are Gaussian. If necessary, the analysis might resort to nonparametric analysis such as Wilcoxon rank sum test. In the case of continuous endpoints, the mean value, standard deviation, median, interquartile distance, and minimum and maximum for both groups will be indicated separately and in total. For categorical endpoints, absolute and relative frequencies will be given for the two groups separately and in total. Survival (Overall survival and disease-specific survival) as well as time to event analyses (Occurrence of AEs) will be evaluated using univariate and multivariate Cox proportional hazard models. Kaplan-Meier plots will be presented and allow to assess whether the proportional hazard assumption is violated. In case of a violation, we will use time-axis division by dividing the time to event into time-intervals that fulfill the proportionality assumption. Univariate and multivariate logistic regression models might be used for binary secondary endpoints including propensity score matching if necessary. Descriptive p-values will be given together with 95% confidence intervals. In the EUS-BD group, descriptive p-values will also be shown by the method.

## Ethical considerations and declarations

The PUMa trial is being performed in accordance with the Declaration of Helsinki. Written consent will be obtained from each study participant. The Institutional Review Board II of the University of Heidelberg (Germany) approved the complete study protocol (S2 File) on the 6[th] of April 2018 (2018-522N-MA) (S3 File), including the last amendment on the 13[th] of August 2019 (S4 File). Local ethics approval was obtained from all further 15 participating hospitals in Spain and in Germany. The PUMa trial was registered with the identification number NCT03546049 in ClinicalTrials.gov on the 22[nd] of May 2018.

## Status and timeline of the study

The first patient was enrolled on the 28th of December 2018. To date (5th of December 2021), 93/216 (43%) patients have been recruited, and further inclusion is on schedule. Total duration: 56 months, Duration of clinical phase: 50 months, FSI (first subject in): 28.12.2018, LSI (last subject in): 28.12.2023, LSO (last subject out): 31.5.2024, DBL (database lock): 30.06.2024, Completion of statistical analysis: 01.10.2024, Completion of study report: 01.12.2024.

# Discussion

## Strength of this study

The strength of this study is the prospective multicenter international design comparing the intervention of PTBD with EUS-BD in a clearly defined entity of malignant unresectable distal bile duct obstruction. The target case number is the largest so far. EUS-BD comprises all three relevant techniques: EUS-CDS, EUS-HGS and EUS-AGS. Both interventions are compared "at eye level" using the currently best method of biliary drainage when ERCP has failed. Unique is also the detailed documentation of AEs according to a recommended severity grading system as both interventions can be associated with many different AEs. PTBD can be associated with severe pain. Therefore, patient reported pain will be additionally documented before and after PTBD and EUS-BD. This study might help to clarify whether PTBD is non-inferior to EUS-BD concerning technical success, and whether one of both interventions is superior in terms of efficacy and safety in one or more secondary endpoints.

## Limitations

Non-randomization might be considered as the main limitation of this study. Non-randomization was the consciously chosen design for this study. As mentioned above both PTBD as well as EUS-BD are complex and less frequent "second-line" biliary drainage procedures in distal malignant bile duct obstruction even in high volume centers. Therefore, centers usually prefer only one of both interventions to deliver the highest quality of treatment for their patients. Vice versa, randomization would cause a bias if one center would be obliged to offer one of both biliary drainage procedures in lower quality than the other biliary drainage procedure [21]. As the historically "younger" EUS-BD is increasingly implemented in interventional gastrointestinal endoscopy centers for other indications, EUS-BD and PTBD might be offered with the same high quality in the identical center in near future. EUS-guided rendez vous (EUS-RV) was not included in the EUS-BD study arm, as the main part of the procedure is performed as ERCP, and AEs might be associated more with ERCP than with EUS-BD. Only Departments of Gastroenterology are participating in this multicenter study although interventional radiologists perform PTBD as well, whereas EUS-BD is usually exclusively performed by interventional endoscopists [22]. Therefore, results of this study might not be applied on PTBDs performed by interventional radiologists. And finally, follow up-analysis of the treated patients might be confounded by the underlying tumor entity and the received palliative medical oncological treatment. This must be kept in mind when calculating and comparing the re-intervention rate, DSS and OAS.

## Dissemination plans

There is a commitment to publish the results of this study in an international journal which is listed in the Journal Citation Report. No patient names will appear in any article, and no one, except for the researchers in this study and the members of the local Institutional Review

Boards, will have access to the data, in accordance with the Law on the Protection of Data of a Personal Nature.

## Amendments

Amendments to the protocol will be reported to all investigators, the local Institutional Review Board, all participants, and the journal.

## Premature termination of the study

An interim analysis will be performed when 100 and 150 patients are recruited. If an intervention can be proven to be clearly beneficial or harmful compared to the concurrent control based on a pre-defined analysis of an incomplete data set while the study is on-going, the investigators may stop the study early. Every patient has the right to withdraw his or her consent to participate in this clinical trial at any time, without providing a reason. If possible, the time and reason for the termination of the study will be recorded in the CRF. The data collected until then will also be noted in the CRF. Reasons for premature termination of the study may be the withdrawal of consent by the patient or if the study participant can no longer be contacted. In the case of an accumulation of SUEs in a trial center or trial arm, the Steering Committee will be involved. The Steering Committee reserves the right to draw the appropriate consequences from the reports received, such as the closure of a trial center or the termination of the study. All investigators involved must be informed immediately of any stoppage or interruption of the clinical trial. The decision is binding among all study centers and investigators. If the clinical trial is terminated prematurely, all study materials (completed, partially completed) must be returned to data management.

## Supporting information

**S1 File. List of study centers.**
(DOCX)

**S2 File. Complete study protocol.**
(DOCX)

**S3 File. Approvals of the local ethics committee of the leading study center.**
(DOCX)

**S4 File. Amendments to the study protocol.**
(DOCX)

**S5 File. SPIRIT checklist.**
(DOC)

## Acknowledgments

Special thanks go to Prof. Dr. med. Uwe Will (Gera/Germany) who generously supported the development of the study concept and design with his comprehensive expertise in EUS-guided interventions. Furthermore, special thanks go to María Dolores Gómez Bernal who created the illustrations of the biliary drainage procedures.

## Author Contributions

**Conceptualization:** Daniel Schmitz, Arthur Schmidt.

**Data curation:** Carlos T. Valiente, Francisco Javier García-Alonso, Amaia Arrubla, Tobias Kleemann.

**Investigation:** Daniel Schmitz, Markus Dollhopf, Manuel Perez-Miranda, Armin Küllmer, Joan Gornals, Juan Vila, Jochen Weigt, Torsten Voigtländer, Eduardo Redondo-Cerezo, Thomas von Hahn, Jörg Albert, Stephan vom Dahl, Torsten Beyna, Dirk Hartmann, Franziska Franck, Francisco Javier García-Alonso, Arthur Schmidt, Albert Garcia-Sumalla, Amaia Arrubla, Markus Joerdens, Tobias Kleemann, José Ramón Aparicio Tomo, Jochen Rudi.

**Methodology:** Felix Grassmann.

**Project administration:** Daniel Schmitz, Amaia Arrubla.

**Resources:** Franziska Franck, Tobias Kleemann.

**Supervision:** Jochen Weigt, Jochen Rudi.

**Validation:** Arthur Schmidt.

**Visualization:** Daniel Schmitz.

**Writing – original draft:** Daniel Schmitz, Carlos T. Valiente.

**Writing – review & editing:** Joan Gornals, Juan Vila.

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
