## [Decision Letter · Decision Letter 0]

13 Jun 2022

PONE-D-21-38739

Percutaneous transhepatic or endoscopic ultrasound-guided biliary drainage in malignant distal bile duct obstruction using a self-expanding metal stent: Study protocol for a prospective European multicenter trial (PUMa trial)

PLOS ONE

Dear Dr. Schmitz,

Thank you for submitting your manuscript to PLOS ONE. After careful consideration, we feel that it has merit but does not fully meet PLOS ONE’s publication criteria as it currently stands. Therefore, we invite you to submit a revised version of the manuscript that addresses the points raised during the review process.

The manuscript has been evaluated by three reviewers, and their comments are available below.<o:p></o:p>

The reviewers have raised a number of major concerns. They feel that the limitations of the study should be better described, and they also note concerns about the statistical analyses presented and request re-analyses be completed.<o:p></o:p>

Could you please carefully revise the manuscript to address all comments raised?<o:p></o:p>

We look forward to receiving your revised manuscript.

Kind regards,

Lorena Verduci

Staff Editor

PLOS ONE

**Journal requirements:**

2. We note that the original protocol file you uploaded contains a confidentiality notice indicating that the protocol may not be shared publicly or be published. Please note, however, that the PLOS Editorial Policy requires that the original protocol be published alongside your manuscript in the event of acceptance. Please note that should your paper be accepted, all content including the protocol will be published under the Creative Commons Attribution (CC BY) 4.0 license, which means that it will be freely available online, and any third party is permitted to access, download, copy, distribute, and use these materials in any way, even commercially, with proper attribution.

Therefore, we ask that you please seek permission from the study sponsor or body imposing the restriction on sharing this document to publish this protocol under CC BY 4.0 if your work is accepted. We kindly ask that you upload a formal statement signed by an institutional representative clarifying whether you will be able to comply with this policy. Additionally, please upload a clean copy of the protocol with the confidentiality notice (and any copyrighted institutional logos or signatures) removed.

4. Please upload a copy of Figure 4, to which you refer in your text on page 24. If the figure is no longer to be included as part of the submission please remove all reference to it within the text.

Reviewers' comments:

Reviewer's Responses to Questions

**Comments to the Author**

1. Does the manuscript provide a valid rationale for the proposed study, with clearly identified and justified research questions?

Reviewer #1: Yes

Reviewer #2: Yes

Reviewer #3: Yes

2. Is the protocol technically sound and planned in a manner that will lead to a meaningful outcome and allow testing the stated hypotheses?

Reviewer #1: Yes

Reviewer #2: Yes

Reviewer #3: Yes

3. Is the methodology feasible and described in sufficient detail to allow the work to be replicable?

Reviewer #1: Yes

Reviewer #2: Yes

Reviewer #3: Yes

4. Have the authors described where all data underlying the findings will be made available when the study is complete?

Reviewer #1: Yes

Reviewer #2: Yes

Reviewer #3: Yes

5. Is the manuscript presented in an intelligible fashion and written in standard English?

Reviewer #1: Yes

Reviewer #2: Yes

Reviewer #3: Yes

6. Review Comments to the Author

You may also provide optional suggestions and comments to authors that they might find helpful in planning their study.

Reviewer #1: I enjoyed reading the study protocol, which proposes a prospective, multicenter, non-randomized 2-arm non-inferiority study. Sample size and power justifications are adequate. my comments are as follows:

1. Comments like "appropriate statistical methods ...." (see Statistical analysis section) should be avoided; clearly specify that you maybe using 2-group t-tests for variables that are Gaussian, and may resort to nonparametric analysis (such as, Wilcoxon rank sum test)

2. Since this will be a non-randomized design, what will be the corresponding CONSORT statements/guidelines to present? That statement is missing in this STudy Protocol

3. Clearly, owing to the multi-center nature of the proposal, I was wondering why the authors didn't include analysis plan that "controls" for this center clustering effect.

4. Survival analysis plan includes very basic assessments using Kaplan-Meier plots. Why isn't a regression plan (say, via Cox models, or alternative models when PH assumption is not satisfied) not part of the protocol, in the event that a variety of other covariables can be collected during the study?

Reviewer #2: This is an interesting multicenter prospective study in Europe. As the second-line treatment for biliary obstruction, EUS-BD and PTBD are raised as candidates. Therefore, this study should provide an important evidence for clinical practices for biliary diseases. However, I have a concern on it.

Major

1. As authors mention, non-randomized study is the limitation in this study. I recommend authors to describe how to overcome this limitation. For example, propensity score matching or logistic regression analysis can be employed.

Reviewer #3: This paper described a study protocol comparing the intervention of PTBD with EUS-BD in a prospective design. This study is well organized and well written, however, there are some concerns below.

1)Being biased background since this study will include gastric cancers and hepatobiliary malignancy that developed distal bile duct obstruction, it should be mentioned as one of the limitations of this study. Biological behavior of gastric cancers is quite different from that of hepato-biliary cancers.

2)Concerning the location of malignant biliary obstruction, please define the distal biliary obstruction for reader to understand. Would it be difficult to make a distinction difference between the distal and proximal?

7. PLOS authors have the option to publish the peer review history of their article (what does this mean?). If published, this will include your full peer review and any attached files.

Reviewer #1: No

Reviewer #2: No

Reviewer #3: No

---

## [Author Response · Author response to Decision Letter 0]

25 Jul 2022

July 25, 2022

Response to reviewers’ comments on manuscript PONE-D-21-38739

Dear Lorena Verduci, Staff Editor of PLOS ONE,

thank you that our manuscript PONE-D-21-38739 was commented by your reviewers and considered for revision.

The revised manuscript with track changes uses blue for reviewer 1, red for reviewer 2 and green for reviewer 3.

In the following, we would like to respond to the reviewers´ comments in detail. 

Response to the requested journal requirements

1. The missing PLOS ONE's style requirements were added and the whole manuscript corrected

2. The study investigators decided to omit the sentence “This protocol, the CRFs, and other study documents and materials must be treated with the highest degree of confidentiality and may not be disclosed to third parties unless the principal gives his express consent. Employees of the investigator are also bound by this agreement” in the original protocol. Furthermore, figure 1 was omitted as this is a figure with copyright (courtesy from Prof. Will). The cleaned protocol is uploaded as S2_Complete Study Protocol

3. A redundant ethics statement was deleted in the section “Amendments”

4. The reference to Figure 4 was deleted as no longer needed

Response to the specific reviewers‘ comments:

Reviewer 1: 

1. Comments like "appropriate statistical methods ...." (see Statistical analysis section) should be avoided; clearly specify that you maybe using 2-group t-tests for variables that are Gaussian, and may resort to nonparametric analysis (such as, Wilcoxon rank sum test)

Answer: The sentence in the Statistical Analysis section was specified to: 

Two group t-tests might be used for variables that are Gaussian. If necessary, the analysis might resort to nonparametric analysis such as Wilcoxon rank sum test

(Line 377-379)

2. Since this will be a non-randomized design, what will be the corresponding 

 CONSORT statements/guidelines to present? That statement is missing in this Study 

 Protocol

 Answer: Reasons for the study design as a non-randomized study are given in the 

 section “Limitations”. Inclusion process will be reported in an adapted 

 CONSORT flow diagram as specified in the section “Follow up”. The sentence was 

 revised to:

 Excluded patients and reasons for exclusion will be presented in an adapted 

 CONSORT flow diagram [19] (Line 273-274).

3. Clearly, owing to the multi-center nature of the proposal, I was wondering why the authors didn't include analysis plan that "controls" for this center clustering effect.

Answer:

The concept of a non-randomized prospective study was chosen for this research to avoid a center clustering effect. However, logistic regression models might be used in the statistical analysis if necessary.

4. Survival analysis plan includes very basic assessments using Kaplan-Meier plots. Why isn't a regression plan (say, via Cox models, or alternative models when PH assumption is not satisfied) not part of the protocol, if a variety of other covariables can be collected during the study?

 Answer:

 Text was changed to: 

 Logistic regression models might be used for secondary endpoints including 

 propensity score matching if necessary (Line 384-386).

Reviewer 2: 

1. As authors mention, non-randomized study is the limitation in this study. I recommend authors to describe how to overcome this limitation. For example, propensity score matching, or logistic regression analysis can be employed.

 Answer:

 Text was changed to: 

 Logistic regression models might be used for secondary endpoints including 

 propensity score matching if necessary (Line 384-386)

Reviewer 3: 

1. Being biased background since this study will include gastric cancers and hepatobiliary malignancy that developed distal bile duct obstruction, it should be mentioned as one of the limitations of this study. Biological behavior of gastric cancers is quite different from that of hepato-biliary cancers.

 Answer: 

 We agree to this comment. Survival after successful technical and clinical biliary 

 drainage will be affected by tumor entity and other confounding factors such as 

 medical oncological therapy. Therefore, this issue was added to the section 

 “Limitations”:

 And finally, follow up-analysis of the treated patients might be confounded by the 

 underlying tumor entity and the received palliative medical oncological treatment. 

 (line 434-436)

2. Concerning the location of malignant biliary obstruction, please define the distal biliary obstruction for reader to understand. Would it be difficult to make a distinction difference between the distal and proximal?

Answer:

We specified the distinction between proximal and distal extrahepatic bile duct obstruction by adding a further item to the exclusion criteria:

 Proximal malignant extrahepatic bile duct obstruction defined as obstruction at the 

 level of the bifurcation of the extrahepatic bile duct and proximal of it (line 212-213)

With kind regards,

Daniel Schmitz

PS: Please use the following email address for future correspondence: Daniel.Schmitz@helios-gesundheit.de

---

## [Decision Letter · Decision Letter 1]

11 Aug 2022

PONE-D-21-38739R1Percutaneous transhepatic or endoscopic ultrasound-guided biliary drainage in malignant distal bile duct obstruction using a self-expanding metal stent: Study protocol for a prospective European multicenter trial (PUMa trial)PLOS ONE

Dear Dr. Schmitz,

Thank you for submitting your manuscript to PLOS ONE. After careful consideration, we feel that it has merit but does not fully meet PLOS ONE’s publication criteria as it currently stands. Therefore, we invite you to submit a revised version of the manuscript that addresses the points raised during the review process.

Specifically, please provide your response to the reviewer's remaining concern.

We look forward to receiving your revised manuscript.

Kind regards,

Jianhong Zhou

Staff Editor

PLOS ONE

Journal Requirements:

Reviewers' comments:

Reviewer's Responses to Questions

**Comments to the Author**

1. Does the manuscript provide a valid rationale for the proposed study, with clearly identified and justified research questions?

Reviewer #1: Partly

Reviewer #2: Yes

Reviewer #3: Yes

2. Is the protocol technically sound and planned in a manner that will lead to a meaningful outcome and allow testing the stated hypotheses?

Reviewer #1: Partly

Reviewer #2: Yes

Reviewer #3: Yes

3. Is the methodology feasible and described in sufficient detail to allow the work to be replicable?

Reviewer #1: Yes

Reviewer #2: Yes

Reviewer #3: Yes

4. Have the authors described where all data underlying the findings will be made available when the study is complete?

Reviewer #1: No

Reviewer #2: Yes

Reviewer #3: Yes

5. Is the manuscript presented in an intelligible fashion and written in standard English?

Reviewer #1: Yes

Reviewer #2: Yes

Reviewer #3: Yes

6. Review Comments to the Author

You may also provide optional suggestions and comments to authors that they might find helpful in planning their study.

Reviewer #1: The authors addressed my questions, except one.

I asked: Survival analysis plan includes very basic assessments using Kaplan-Meier plots.

Why isn't a regression plan (say, via Cox models, or alternative models when PH

assumption is not satisfied) not part of the protocol, if a variety of other covariables can

be collected during the study?

Authors replied: Text was changed to:

Logistic regression models might be used for secondary endpoints including

propensity score matching if necessary (Line 384-386).

I have no clue what the authors meant here. If one has survival/time-to-event endpoints, one would like to do a Cox regression; logistic regression is a separate stuff. Please clarify, or make the appropriate correction.

Reviewer #2: Authors properly revised the manuscript according to my comments. I do not have any additional comments on it.

Reviewer #3: I have carefully read the revised version of this paper. The authors perfectly responded to all my concerns. I have no further comments.

7. PLOS authors have the option to publish the peer review history of their article (what does this mean?). If published, this will include your full peer review and any attached files.

Reviewer #1: No

Reviewer #2: No

Reviewer #3: **Yes: **Kosuke Minaga

---

## [Author Response · Author response to Decision Letter 1]

29 Aug 2022

29 August 2022

PONE-D-21-38739R1

Percutaneous transhepatic or endoscopic ultrasound-guided biliary drainage in malignant distal bile duct obstruction using a self-expanding metal stent: Study protocol for a prospective European multicenter trial (PUMa trial)

PLOS ONE

Response to reviewers

I. Journal Requirements:

The reference list was reviewed to ensure that it is complete and correct. All cited papers are complete and correct. 

Felix Grassmann from the Institute of Medical Statistics and Epidemiology was newly added to the author list as he substantially contributed to the statistical concept of the study protocol

II. Response to Reviewers' comments:

Have the authors described where all data underlying the findings will be made available when the study is complete?

Reviewer #1: No

Reviewer #2: Yes

Reviewer #3: Yes

Response: 

The complete study protocol and all study results will be available in the public online repository (FAIRsharing ID 4350; https://fairsharing.org). The manuscript text was changed on page 4 and 13.

Review Comments to the Author

Reviewer #1: The authors addressed my questions, except one.

I asked: Survival analysis plan includes very basic assessments using Kaplan-Meier plots.

Why isn't a regression plan (say, via Cox models, or alternative models when PH

assumption is not satisfied) not part of the protocol, if a variety of other covariables can

be collected during the study?

Authors replied: Text was changed to:

Logistic regression models might be used for secondary endpoints including

propensity score matching if necessary (Line 384-386).

I have no clue what the authors meant here. If one has survival/time-to-event endpoints, one would like to do a Cox regression; logistic regression is a separate stuff. Please clarify or make the appropriate correction.

Response: 

We thank the reviewer for this comment. We have changed the section and now state that we will fit Cox proportional hazard models to analyze survival/time to event analyses. In addition, Kaplan Meier plots will be presented to identify potential violations of the proportional hazard assumptions with potential next steps in case such violations are found.

Manuscript was changed on page 17 (lines 386-393):

Survival (Overall survival and disease-specific survival) as well as time to event analyses (Occurrence of AEs) will be evaluated using univariate and multivariate Cox proportional hazard models. Kaplan-Meier plots will be presented and allow to assess whether the proportional hazard assumption is violated. In case of a violation, we will use time-axis division by dividing the time to event into time-intervals that fulfill the proportionality assumption. Univariate and multivariate logistic regression models might be used for binary secondary endpoints including propensity score matching if necessary.

---

## [Decision Letter · Decision Letter 2]

9 Sep 2022

Percutaneous transhepatic or endoscopic ultrasound-guided biliary drainage in malignant distal bile duct obstruction using a self-expanding metal stent: Study protocol for a prospective European multicenter trial (PUMa trial)

PONE-D-21-38739R2

Dear Dr. Schmitz,

We’re pleased to inform you that your manuscript has been judged scientifically suitable for publication and will be formally accepted for publication once it meets all outstanding technical requirements.

Kind regards,

Jianhong Zhou

Staff Editor

PLOS ONE

Additional Editor Comments (optional):

Reviewers' comments:

Reviewer's Responses to Questions

**Comments to the Author**

1. Does the manuscript provide a valid rationale for the proposed study, with clearly identified and justified research questions?

Reviewer #1: Yes

2. Is the protocol technically sound and planned in a manner that will lead to a meaningful outcome and allow testing the stated hypotheses?

Reviewer #1: Yes

3. Is the methodology feasible and described in sufficient detail to allow the work to be replicable?

Reviewer #1: Yes

4. Have the authors described where all data underlying the findings will be made available when the study is complete?

Reviewer #1: Yes

5. Is the manuscript presented in an intelligible fashion and written in standard English?

Reviewer #1: Yes

6. Review Comments to the Author

You may also provide optional suggestions and comments to authors that they might find helpful in planning their study.

Reviewer #1: The authors were able to address my previous questions, and points of clarifications. I have no further questions for them.

7. PLOS authors have the option to publish the peer review history of their article (what does this mean?). If published, this will include your full peer review and any attached files.

Reviewer #1: No

---

## [Editor Report · Acceptance letter]

20 Oct 2022

PONE-D-21-38739R2 

Percutaneous transhepatic or endoscopic ultrasound-guided biliary drainage in malignant distal bile duct obstruction using a self-expanding metal stent: Study protocol for a prospective European multicenter trial (PUMa trial) 

Dear Dr. Schmitz:

I'm pleased to inform you that your manuscript has been deemed suitable for publication in PLOS ONE. Congratulations! Your manuscript is now with our production department. 

Kind regards, 

on behalf of

Jianhong Zhou 

Staff Editor

PLOS ONE